# Bootstrap Your Own Skills: Learning to Solve New Tasks with Large Language Model Guidance

**Jesse Zhang[1], Jiahui Zhang[1], Karl Pertsch[1], Ziyi Liu[1],**
**Xiang Ren[1]**, **Minsuk Chang[2]**, **Shao-Hua Sun[3]**, **Joseph J. Lim[4]**
[1]University of Southern California, [2]Google AI, [3]National Taiwan University, [4]KAIST
jessez@usc.edu

**Abstract:** We propose BOSS, an approach that automatically learns to solve new long-horizon, complex, and meaningful tasks by growing a learned skill library with minimal supervision. Prior work in reinforcement learning requires expert supervision, in the form of demonstrations or rich reward functions, to learn long-horizon tasks. Instead, our approach BOSS (**BO**otstrapping your own **S**kill**S**) learns to accomplish new tasks by performing "skill bootstrapping," where an agent with a set of primitive skills interacts with the environment to practice new skills without receiving reward feedback for tasks outside of the initial skill set. This bootstrapping phase is guided by large language models (LLMs) that inform the agent of meaningful skills to chain together. Through this process, BOSS builds a wide range of complex and useful behaviors from a basic set of primitive skills. We demonstrate through experiments in realistic household environments that agents trained with our LLM-guided bootstrapping procedure outperform those trained with naïve bootstrapping as well as prior unsupervised skill acquisition methods on zero-shot execution of unseen, long-horizon tasks in new environments. View website at clvrai.com/boss.

## 1 Introduction

Robot learning aims to equip robots with the capability of learning and adapting to novel scenarios. Popular learning approaches like reinforcement learning (RL) excel at learning short-horizon tasks such as pick-and-place [1, 2, 3], but they require dense supervision (e.g., demonstrations [4, 5, 6, 7] or frequent reward feedback [8, 9, 10]) to acquire long-horizon skills.

In contrast, humans can learn complex tasks with much less supervision—take, for example, the process of learning to play tennis: we may initially practice individual skills like forehand and backhand returns under close supervision of a coach, analogous to RL agents practicing simple pick-place skills using demonstrations or dense rewards. Yet importantly, in between coaching sessions, tennis players return to the tennis court and *practice* to combine the acquired basic skills into long-horizon gameplay without supervision from the coach. This allows them to develop a rich repertoire of tennis-playing skills independently and perform better during their next match.

Can we enable agents to similarly practice and expand their skills without close human supervision? We introduce BOSS (**BO**otstrapping your own **S**kill**S**), a framework for learning a rich repertoire of long-horizon skills with minimal human supervision (see Figure 1). Starting from a base set of acquired primitive skills, BOSS performs a *skill bootstrapping phase* in which it progressively grows its skill repertoire by practicing to chain skills into longer-horizon behaviors. BOSS enables us to train generalist agents, starting from a repertoire of only tens of skills, to perform hundreds of long-horizon tasks without additional human supervision.

A crucial question during practice is which skills are meaningful to chain together: randomly chaining tennis moves does not lead to meaningful gameplay; similarly, random chains of pick-place movements do not solve meaningful household tasks. Thus, in BOSS we propose to leverage the rich knowledge captured in large language models (LLMs) to guide skill chaining: given the chain

7th Conference on Robot Learning (CoRL 2023), Atlanta, USA.

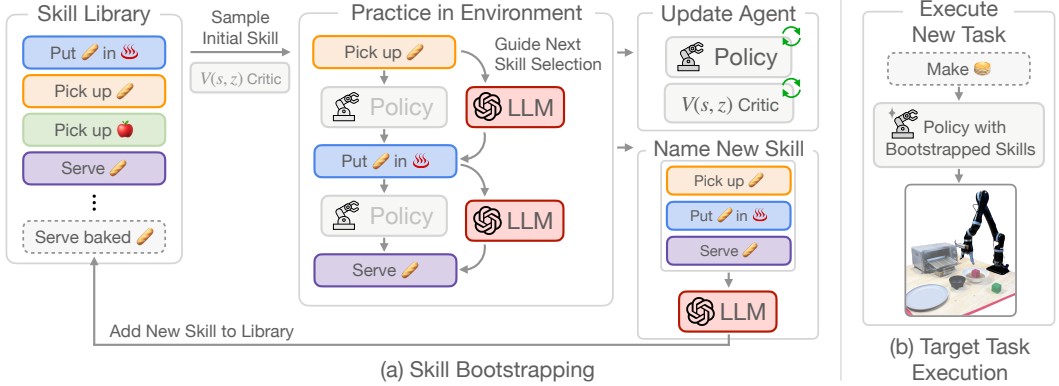

Figure 1: BOSS learns to execute a large set of useful, long-horizon skills with minimal supervision by performing LLM-guided *skill bootstrapping*. **(a):** The agent starts with an initial skill library. During bootstrapping, it practices chaining skills into new long-horizon behaviors using guidance from an LLM. The collected experience is used to update the policy. Newly discovered skill chains are summarized with an LLM and added as new skills into the library for further bootstrapping. Thus, the agent's skill repertoire grows over time. **(b):** After bootstrapping, we condition the policy on novel instructions and show execution in the environment using the bootstrapped skill repertoire.

of executed skills so far, the LLM predicts a distribution over meaningful next skills to sample. Importantly, in contrast to existing approaches that leverage the knowledge captured in LLMs for long-horizon task planning [11, 12, 13, 14], BOSS can use unsupervised environment interactions to *practice* how to chain skills into long-horizon task executions; this practice is crucial especially if the target environment differs from the ones used to train the base skill set. This results in a more robust policy that can compensate for accumulating errors from the initial skill repertoire.

We validate the effectiveness of our proposed approach in simulated household environments from the ALFRED benchmark and on a real robot. Experimental results demonstrate that BOSS can practice effectively with LLM guidance, allowing it to solve long-horizon household tasks in novel environments which prior LLM-based planning and unsupervised exploration approaches fail at.

## 2 Preliminaries and Related Work

**Reinforcement Learning** Reinforcement learning (RL) algorithms aim to learn a policy $\pi(a|s)$ that maximizes the expected discounted return $\mathbb{E}_{a\sim\pi,P}\left[\sum_t \gamma^t R(s_t, a_t, s_{t+1})\right]$ in a Markov Decision Process $\mathcal{M} = (\mathcal{S}, \mathcal{A}, P, \mathcal{R}, \gamma)$, where $\mathcal{S}$ and $\mathcal{A}$ are state and action spaces, $P : \mathcal{S} \times \mathcal{A} \times \mathcal{S} \to \mathbb{R}_+$ represents the transition probability distribution, $\mathcal{R} : \mathcal{S} \times \mathcal{A} \times \mathcal{S} \to \mathbb{R}$ denotes the reward function, and $\gamma$ is the discount factor. Temporal-difference algorithms are a class of RL algorithms that also learn critic functions, denoted $V^\pi(s)$ or $Q^\pi(s, a)$, which represent future discounted returns when following the policy at state $s$ or after taking action $a$ from state $s$, respectively [15]. Standard RL algorithms struggle with learning long-horizon tasks and can be prohibitively sample-inefficient.

**Skill-based RL** To solve long-horizon tasks, prior works have focused on pre-training *skills*, short-horizon behaviors that can be re-combined into long-horizon behaviors [9, 16, 17, 18, 19]. These skills can be represented as learned options [16, 18], sub-goal setting and reaching policies [20, 21], a set of discrete policies [22, 23], or continuous latent spaces that represent behaviors [9, 10, 24, 25, 26]. Yet, most of these approaches need expert supervision (e.g., demonstrations [4, 5, 6, 7, 20, 21, 27], frequent reward feedback [9, 10, 23]). In contrast, BOSS learns to execute long-horizon tasks with minimal human supervision via skill bootstrapping.

**Unsupervised RL** To learn skills without human supervision, recent works have introduced many unsupervised RL objectives, e.g., based on curiosity [28], contrallability [29, 30], and behavior or state diversification [31, 32, 33, 34, 35]. Because these works learn skills from scratch and explore without supervision, they generally focus on locomotion tasks where most behaviors agents can ex-

plore, such as different running gaits, are already meaningful. Few works demonstrate learning of manipulation tasks, but either require hand-crafted state or action spaces [28] or remain constrained to learning simple, short-horizon skills [36, 37]. BOSS makes two improvements to enable bootstrapping of long-horizon tasks: (1) We start from a base repertoire of language-conditioned skills to enable coherent, long-horizon exploration. (2) We leverage an LLM to guide exploration towards meaningful skill-chains within the exponential number of possible long-horizon behaviors.

**Language in RL**  Prior works have employed language to parameterize rich skill sets to train multi-task RL agents [38, 39, 40, 41, 42, 43]. Recent progress in training LLMs has enabled approaches that combine LLMs with pre-trained language-conditioned policies to perform open-loop planning over pre-trained skills [11, 12, 13, 14, 44]. These works do not perform any policy training or finetuning when planning with the LLMs; but instead use the LLMs as top-down planners whose plans are given to fixed low-level skill policies to execute. In contrast, BOSS *pratices* chaining behaviors in the environment during skill bootstrapping and thus learns a more robust, closed-loop policy. This leads to substantially higher success rate for executing long-horizon tasks.

ELLM [45], LMA3 [46], and IMAGINE [47] are closest to our work. ELLM and LMA3 both use an LLM to generate tasks, with the former requiring a captioning model to reward agents and the latter additionally using the LLM to hindsight label past agent trajectories for task completion; instead, we expand upon a learned skill repertoire, allowing for building skill chains while automatically rewarding the agent based on the completion of skills in the chain. Meanwhile, IMAGINE uses language guidance to generate exploration goals, requiring a "social partner" that modifies the environment according to desired goals. In realistic settings, this social partner requires extensive human effort to design. BOSS instead utilizes LLMs to propose goals in a target environment automatically.

# 3 Method

Our method, BOSS (**BO**otstrapping your own **S**kill**S**), automatically learns to solve new long-horizon, complex tasks by growing a learned skill library with minimal supervision. BOSS consists of two phases: (1) it acquires a base repertoire of skills (Section 3.1) and then (2) it practices chaining these skills into long-horizon behaviors in the *skill bootstrapping phase* (Section 3.2). BOSS can then *zero-shot execute* novel natural language instructions describing complex long-horizon tasks.

## 3.1 Pre-training a Language-Conditioned Skill Policy

We assume access to a dataset $\mathcal{D}^L = \{\tau_{z_1}, \tau_{z_2}, \tau_{z_3}, ..., \}$ where $\tau_{z_i}$ denotes a trajectory of $(s, a, s', r)$ tuples and $z_i$ is a freeform language description of the trajectory. We also assume access to a sparse reward function for the primitive skills, e.g., an object detector that can detect if an object is placed in the correct location. For example, if $\tau_{z_i}$ demonstrates a robot arm picking up a mug, then $z_i =$ *"pick up the mug."* and $r = 1$ in the final transition in which the mug is picked up and 0 otherwise. To obtain a language-conditioned primitive skill policy, we train a standard offline RL algorithm on $\mathcal{D}^L$. In our experiments, we use Implicit Q-Learning (IQL) [48] as it is performant and amenable to online fine-tuning. We condition the policy and critic networks on the trajectory's natural language annotation $z$, yielding a language-conditioned policy $\pi(a|s, z)$ and a critic function $V(s, z)$.

## 3.2 Skill Bootstrapping

After learning the language-conditioned primitive skill policy, we perform skill bootstrapping — the agent practices by interacting with the environment, trying new skill chains, then adding them back into its skill repertoire for further bootstrapping. As a result, the agent learns increasingly long-horizon skills without requiring additional supervision beyond the initial set of skills.

**Sampling initial skills.** At the start of bootstrapping, the skill repertoire $Z = \{z_1, z_2, ...\}$ is initialized to the set of pre-trained base skills. Upon initializing the agent in the environment at state $s_1$, we must sample an initial skill. Intuitively, the skill we choose should be executable from $s_1$ i.e., have a high chance of success. Therefore, in every bootstrapping episode, we sample the initial skill according to probabilities generated from the pre-trained value function, $V(s_1, z)$. We then try to execute the sampled skill until a timeout threshold is reached.

**Guiding Skill Chaining via LLMs.** If the first skill execution succeeds, the next step is constructing a longer-horizon behavior by chaining together the first skill with a sampled next skill. Naïvely

choosing the next skill by, for example, sampling at random will likely result in a behavior that is not useful for downstream tasks. Even worse, the likelihood of picking a *bad* skill chain via random sampling increases linearly with the size of the skill repertoire and *exponentially* with the length of the skill chain. For a modestly sized repertoire with 20 skills and a chain length of 5 there are $20^5 = 3.2M$ possible skill chains, only few of which are likely meaningful.

Thus, instead of randomly sampling subsequent skills, we propose to use large language models (LLMs) to guide skill selection. Prior work has demonstrated that modern LLMs capture relevant information about meaningful skill chains [11, 12, 14]. Yet, in contrast to prior *top-down* LLM planning methods, we explore a *bottom-up* approach to learning long-horizon tasks: by allowing our agent to iteratively sample skill chains and practice their execution in the environment, we train more robust long-horizon task policies that achieve higher empirical success rates, particularly when generalizing to unseen environments (see Section 4).

---

**LLM Prompt Example**

Predict the next skill from the following list: Pick up the mug; Turn on the lamp; Put the mug in the coffee machine; ...

1: Pick up the mug.
2:

Figure 2: A shortened LLM prompt. See the full prompt in Appendix A.2.

---

To sample next skills, we prompt the LLM with the current skill repertoire and the chain of skills executed so far. For example, if the agent has just completed *"Pick up the mug"*, we prompt the LLM with the list of skill annotations in $Z$ and then the following prompt: 1. PICK UP THE MUG. 2.____ (see Figure 2). The LLM then *proposes* the next skill by generating text following the prompt. We then map this predicted next skill string back to the set of existing skills in $Z$ by finding the nearest neighbor of $Z$ to the proposed skill annotation in the embedding space of a pre-trained sentence embedding model [49]. To encourage diversity in the practiced skill chains, we repeat this process $N$ times and sample the true next skill from the distribution of LLM-assigned token likelihoods. Finally, if the sampled skill is successfully executed, we repeat the same process for sampling the following skill.[1]

**Learning new skills.** Once an episode concludes, either because a skill times out or because a defined maximum skill chain length is reached, we add the collected data back into the replay buffer with a sparse reward of 1 for every completed skill. For example, if an attempted skill chain contains a total of 3 skills, then the maximum return of the entire trajectory is 3. We then continue policy training via the same offline RL algorithm used to learn the primitive skills—in our case, IQL [48].

Finally, to maximize data efficiency, we relabel the language instructions for the collected episode upon adding it to the replay buffer. Specifically, following prior work [42], we aggregate consecutive skills into *composite* skill instructions using the same LLM as for skill sampling. We then add the composite skill instruction and associated experience to the replay buffer and also add it to our skill repertoire for continued bootstrapping. We store new trajectories with both their lowest level annotations and the LLM-generated composite instructions so the agent can fine-tune its base skills while learning longer-horizon skill chains online. To ensure the agent does not forget its initial skill repertoire, we sample data from the offline dataset $\mathcal{D}^L$ with new data at equal proportions in batch.

In sum, we iterate through these three steps to train a policy during the skill bootstrapping phase: (1) Sampling initial skills using the value function. (2) Sampling next skills by prompting the LLM with skills executed so far. (3) Adding learned skills to the skill library and training on collected agent experience. Algorithm 1 presents a brief overview. The implementation details can be found in Section B and Algorithm 2 in Appendix describes the full algorithm.

---

**Algorithm 1** BOSS Pseudocode.

1: Train policy $\pi$ on initial skill repertoire
2: **for** skill bootstrapping episode **do**
3:     Sample initial skill $z$ and execute
4:     **while** not episode timeout **do**
5:         Sample next skill from LLM and execute
6:     Construct composite skill and add to repertoire
7:     Update policy $\pi$

---

[1]Note that we do not treat invalid LLM skill chain proposals, like asking the agent to "put keys in a safe" when it has not yet picked any keys up, in a special manner. If the proposal is poor, the agent will fail and the value of the skill will drop with training, making it unlikely to sample the skill chain again.

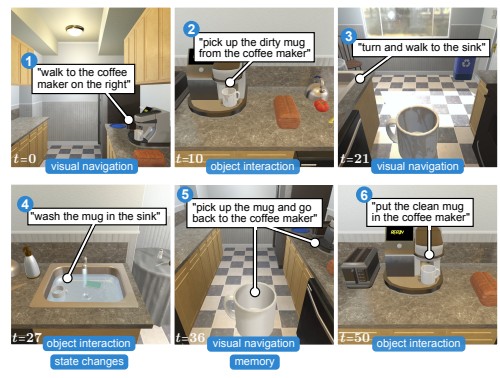

(a) ALFRED benchmark.

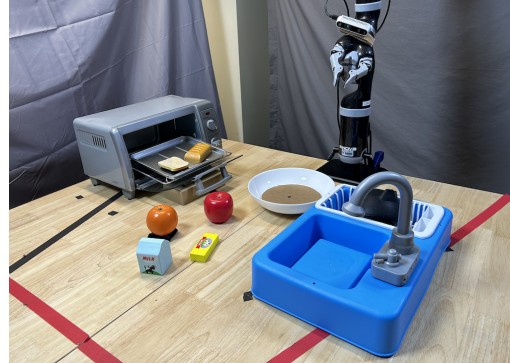

(b) Real world Jaco arm setup.

Figure 3: **Environments.** (a) **The ALFRED environment** is a benchmark for learning agents that can follow natural language instructions to fulfill household tasks. This illustration was drawn from Shridhar et al. [50] with permission. (b) **Real-world Jaco arm:** Our real-world kitchen manipulation tabletop environment based on RGB image inputs.

## 4 Experimental Evaluation

The goal of our experiments is to test BOSS's ability to acquire long-horizon, complex, and meaningful behaviors. We compare to unsupervised RL and zero-shot planning methods in two challenging, image-based control environments: solving household tasks in the ALFRED simulator [50] and kitchen manipulation tasks with a real-world Jaco robot arm. Concretely, we aim to answer the following questions: (1) Can BOSS learn a rich repertoire of useful skills during skill bootstrapping? (2) How do BOSS's acquired skills compare to skills learned by unsupervised RL methods? (3) Can BOSS directly be applied on real robot hardware?

### 4.1 Experimental Setup

**ALFRED Environment.** We test our approach in the ALFRED simulator [50] (see Figure 3a), since its 100+ floorplans with many interactable objects provide a rich environment for learning numerous long-horizon household tasks. We leverage a modified version of the ALFRED simulator [?] that allows for online RL interactions via a gym interface with $300 \times 300$ egocentric RGB image observations. The action space consists of 12 discrete action choices (e.g. turn left, look up, pick up object), along with 82 discrete object types, first proposed by Pashevich et al. [51]. To train the skills in our initial skill library, we leverage the ALFRED dataset of $73k$ primitive skill demonstrations with language instructions. For bootstrapping we use four unseen floorplans. In each floorplan we define 10 evaluation tasks, each of which requires 2 to 8 primitive skills to complete.

**Real-Robot Kitchen Manipulation.** We evaluate our method with a real-robot manipulation setup in which a Kinova Jaco 2 robot arm needs to solve stylized kitchen tasks in a table-top environment (see Figure 3b). The observations consist of concatenated RGB images from a third-person and a wrist-mounted camera. The robot is controlled with continuous end-effector displacements and discrete gripper open/stay/close commands at a frequency of $10Hz$. To train the initial skills, we collect a dataset of $6k$ language-annotated primitive skill demonstrations via human teleoperation. We perform bootstrapping and evaluate the agents in a table setup with unseen object arrangements.

**Training and Evaluation Procedure.** We equip the policy with the initial primitive skill library by training it for 150 epochs on the respective pre-collected demonstration datasets using IQL [48] (see Section 3.1). We then perform 500,000 and 15,000 steps (∼17 min of robot interaction time) of online skill bootstrapping in the respective unseen eval environments of ALFRED and the real robot setup. Note that for ALFRED we train separate agents for each floorplan, mimicking a scenario in which an agent is dropped into a new household and acquires skills with minimal supervision. After bootstrapping, we evaluate the trained agents *zero-shot* on the held-out evaluation tasks by conditioning the policy on the respective language instruction. To perform well in this evaluation setting, an agent needs to acquire a *large* number of *useful* skills during online environment interactions.

**Baselines.** We compare BOSS to prior works that can learn a wide range of skills with minimal supervision: (1) unsupervised RL approaches that, like BOSS, learn from environment interactions without additional feedback and (2) large-language model based planners, that leverage the knowledge captured in large pre-trained language models to "bootstrap" given skill libraries into long-horizon behaviors. Concretely, we are comparing to the following approaches:

- **CIC** [52]: SoTA method on the unsupervised RL benchmark [53], expands its skill library with a contrastive alignment objective during bootstrapping. For fair comparison, we pre-train CIC's policy on the same primitive skill dataset used in BOSS before unsupervised bootstrapping.

- **SayCan** [12]: Leverages a pre-trained LLM to break down a given task into step-by-step instructions, i.e., "primitive skills", by ranking skills from a given library. We implement SayCan using the same primitive skill policy pre-trained via offline RL as in BOSS. We use the same LLM as our method, and adapt SayCan's LLM prompt for our environment. Notably, SayCan and similar LLM planning work have no mechanism for fine-tuning to new environments.

- **SayCan+P**: To evaluate the effects of online bootstrapping vs. top-down LLM planning in isolation, we evaluate a SayCan variant that uses *our* LLM-based *skill proposal* mechanism, which leverages the LLM to *generate* step-by-step instructions in place of SayCan's original skill ranking method. We found this to perform better than standard SayCan in our evaluation.

- **SayCan+PF**: SayCan+P on policies fine-tuned in the target environments for the same number of steps as BOSS by sampling single skills with the value function and learning to execute them. This compares the effect of BOSS *learning to chain* skills in the target environments.

Additionally, we evaluate (1) an **Oracle** that finetunes the pre-trained primitive skill policy directly on the target tasks, serving as an upper bound, and (2) a pre-trained primitive skill policy *without any* bootstrapping (**No Bootstrap**), serving as a performance lower bound.

All methods utilize the same base primitive skill policy pre-trained on the same demonstration data. We implement a transformer policy and critic architecture based on Pashevich et al. [51] trained with the IQL algorithm [48]. All results reported are inter-quartile means and standard deviations over 5 seeds [54]. Finally, Saycan and BOSS all use the LLaMA-13b open-source, 13-billion parameter LLM [55]. For more baseline implementation and training details, see Appendix B.

### 4.2 BOSS Bootstrapping Learns Useful Skills

**ALFRED.** Overall, BOSS achieves superior performance to all non-oracle baselines, with better oracle-normalized return at longer, length 3 and 4 tasks than the best baselines, and BOSS is the **only** method to achieve non-zero success rates across all lengths of tasks. From Table 1, the gap between BOSS and best baselines is largest on the length 4 tasks, indicating the benefit of BOSS' LLM-guided skill bootstrapping in

Table 1: Inter-quartile means (IQMs) and standard deviations of oracle-normalized returns, i.e., number of solved subtasks, broken down by task length, across the ALFRED evaluation tasks. We also report oracle-normalized success rate in the last column. We do not report results for length 6 and 8 tasks since not even the oracle was able to learn these.

| Method | Returns by Evaluation Task Length | | | Average | |
| --- | --- | --- | --- | --- | --- |
| | Length 2 | Length 3 | Length 4 | Return | Success |
| No Bootstrap | 0.03 +- 0.02 | 0.05 +- 0.07 | 0.08 +- 0.09 | 0.03 +- 0.01 | 0.00 +- 0.00 |
| CIC [52] | 0.02 +- 0.02 | 0.25 +- 0.08 | 0.18 +- 0.07 | 0.11 +- 0.01 | 0.00 +- 0.00 |
| SayCan [12] | 0.06 +- 0.02 | 0.14 +- 0.00 | 0.10 +- 0.12 | 0.06 +- 0.00 | 0.00 +- 0.00 |
| SayCan + P | 0.08 +- 0.04 | 0.28 +- 0.00 | 0.20 +- 0.15 | 0.12 +- 0.01 | 0.00 +- 0.00 |
| SayCan + PF | **0.64 +- 0.06** | **0.49 +- 0.20** | 0.59 +- 0.02 | **0.57 +- 0.05** | 0.00 +- 0.00 |
| BOSS (ours) | **0.47 +- 0.12** | **0.59 +- 0.13** | **0.81 +- 0.13** | **0.57 +- 0.06** | **0.57 +- 0.14** |

learning difficult, longer-horizon tasks without task supervision. CIC can make some progress in some length 3 and 4 tasks, but its contrastive objective generally fails to finetune the primitive skills into meaningful long-horizon skills. Saycan+P performs better than Saycan, indicating that our proposal mechanism better extracts a more meaningful distribution of skills from an LLM, but even Saycan+P greatly falls short of BOSS' performance as it is not robust to execution failures incurred from directly using the pre-trained policy in unseen floor plans. Saycan+PF performs better as it first fine-tunes its policies, but it still achieves a 0% success rate compared to BOSS' 57%. Additional analyses we perform in Appendix C.1 demonstrates that in SayCan+P, 95.8% of all unsuccessful SayCan+P trajectories are caused by policy execution failures. SayCan+PF is only slightly better: 95.0% are caused by policy execution failures, indicating that naïve fine-tuning in the target en-

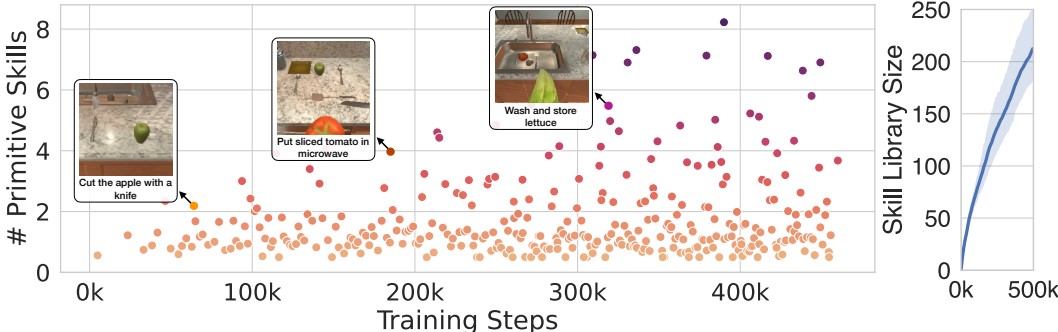

Figure 4: **Left**: The number of subtasks in skills executed *during* skill bootstrapping by BOSS in one of the unseen ALFRED floorplans. BOSS progressively learns longer skill chains throughout the course of training. **Right**: The number of newly acquired skills by BOSS throughout training.

| (1) Pick up the pillow off of the seat of the blue chair
(2) Put the pillow vertically on the couch to the left of the newspaper | (1) Go to the area between the cabinets and the toilet
(2) Pick up the empty toilet paper tube behind the toilet brush
(3) Place the toilet paper tube upright to the left of the full toilet paper roll
(4) Close the cabinet door | (1) Take the apple on the right from the sink
(2) Pick up the knife from the counter
(3) Cut the apple into pieces
(4) Put the apple on the right of the statue and in front of the salt | (1) Pick up the white pencil on the desk
(2) Place the white pencil on the desk near the books
(3) Pick up the books from the bed
(4) Turn on the lamp |
|---|---|---|---|
| Put the pillow on the couch next to the newspaper. | Put the empty toilet paper tube next to the full toilet paper roll. | Cut the apple and put it on the right of the statue. | Place the white pencil on the desk next to the books and then look at the book from the bed under the lamp light. |

Figure 5: Example skill chains (light gray) and new skill summaries (dark grey) learned by BOSS during skill bootstrapping. LLM-guidance ensures meaningful skill chains and summaries.

vironment is ineffective for solving long-horizon tasks. Since BOSS learns to finetune individual primitive skills and transition *between* skills using a closed-loop policy, it performs much better on complex, long-horizon language-specified tasks in unseen environments.

We display qualitative examples of a length 2 and 3 task in appendix Figure 10, where we can see that BOSS successfully completes the tasks whereas Saycan suffers from execution failures, getting stuck while attempting to manipulate objects, and CIC navigates around performing random behaviors (Figure 10a) or gets stuck navigating around objects (Figure 10b). We show qualitative examples of learned skills in Figure 5 and perform additional experiments and analysis in Appendix C.1.

**Real Robot.** In our real world experiments, we compare BOSS to **ProgPrompt** [14], a similar LLM planning method to Saycan that has been extensively evaluated on real-world tabletop robot manipulation environments similar to ours. We also augment it with prompt examples similar to ours and our skill proposal mechanism. Here, we evaluate on 4 tasks, 2 of length 2 and 2 of length 4 after performing bootstrapping. Results in Table 2 demonstrate that both methods perform similarly on length 2 tasks, but only BOSS achieves nonzero success rate on more difficult length 4 tasks as it is able to learn to chain together long-horizon skills in the new environment. See Appendix C.2 for more detailed task information.

Table 2: Success rates, split by task length, across the 4 robot eval tasks in an unseen table arrangement.

|  | Evaluation Task Length | |
|---|---|---|
| Method | Length 2 | Length 4 |
| ProgPrompt [14] | 0.65 +- 0.15 | 0.00 +- 0.00 |
| BOSS (ours) | 0.50 +- 0.30 | **0.15 +- 0.05** |

#### 4.2.1 Ablation Studies

To better analyze the effect of our core contribution, the usage of LLM guidance during skill bootstrapping, we compare to the following variants of our approach:

- **BOSS-OPT1**: BOSS bootstrapping with a weaker 1-billion parameter LLM, OPT-1 [56].

- **BOSS-Rand**: An ablation of our approach BOSS that uses *no* LLM guidance during skill bootstrapping and simply selects the next skill at random from the current skill library.

We report results in Table 3. The analysis shows the importance of accurate LLM guidance during skill bootstrapping for learning useful skills. Using an LLM with lower performance (OPT1) results in degraded overall performance. Yet, bootstrapping without any LLM guidance performs even worse. Interestingly, the performance gap between BOSS and its variants widens for longer task lengths. Intuitively, the longer the task, the more possible other, less useful tasks of the same length could be learned by the agent during bootstrapping. Thus, particularly for long tasks accurate LLM guidance is helpful.

Table 3: ALFRED ablation returns.

| | Evaluation Task Length | | | |
| Method | Length 2 | Length 3 | Length 4 | Average |
| --- | --- | --- | --- | --- |
| BOSS (ours) | 0.47 +- 0.12 | 0.59 +- 0.13 | 0.81 +- 0.13 | **0.57 +- 0.06** |
| BOSS-OPT1 | 0.39 +- 0.08 | 0.36 +- 0.07 | 0.56 +- 0.08 | 0.49 +- 0.07 |
| BOSS-Rand | 0.32 +- 0.03 | 0.29 +- 0.11 | 0.61 +- 0.16 | 0.43 +- 0.06 |

To further analyze this, we compare the sizes of the learned skill libraries between BOSS bootstrapped with LLaMA-13B guidance vs. random skill selection (BOSS-Rand) in Figure 6. Perhaps surprisingly, the random skill chaining ablation learns *more* skills than BOSS – its skill library grows faster during bootstrapping. Yet, Table 3 shows that it has lower performance. This indicates, that while BOSS-Rand learns many skills, it learns less *meaningful* skills. A qualitative analysis supports this intuition: many of the learned skills contain repetitions and meaningless skill chains. This underlines the importance of LLM guidance during skill bootstrapping. Furthermore, the positive correlation between the powerfulness of the used guidance LLM (1B → 13B parameters) and the evaluation task performance suggests that future, even more powerful LLMs can lead to even better skill bootstrapping.

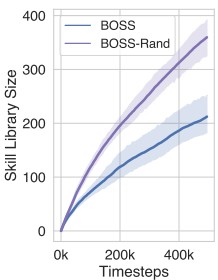

Figure 6: Skill library size during bootstrapping.

## 5  Discussion

We propose BOSS, an approach that learns a diverse set of long-horizon tasks with minimal supervision via LLM-guided skill bootstrapping. Starting from an initial library of skills, BOSS acquires new behaviors by practicing to chain skills while using LLMs to guide skill selection. We demonstrate in a complex household simulator and real robot manipulation tasks that BOSS can learn more useful skills during bootstrapping than prior methods.

**Limitations.** While BOSS learns a large repertoire of skills with minimal supervision, it still has limitations that prevent it from truly fulfilling the vision of agents autonomously acquiring skills in new environments. BOSS requires environment resets between bootstrapping episodes, which are currently performed by a human in our real world experiments. Also, we require success detection for each of the primitive skills during bootstrapping. Future research can investigate using advances in reset-free RL [57, 58] to approach the goal of truly autonomous skill learning. Furthermore, BOSS greedily proposes new skill chains one skill at a time, this greedy skill chaining process may not be optimal for generating consistent long-horizon behaviors beyond a certain length. In future work, we plan to explore mechanisms to propose long-horizon tasks that are broken down to individual skills in conjunction with the greedy skill chaining of BOSS. Finally, BOSS is currently limited to skills that are combinations of skills in its initial skill library. Extending our work with unsupervised RL [59, 52] techniques for learning new *low-level* skills is an exciting direction for future work.

## Acknowledgments

We thank Ishika Singh for her assistance with implementing and debugging ProgPrompt. This work was supported by a USC Viterbi Fellowship, Institute of Information & Communications Technology Planning & Evaluation (IITP) grants (No.2019-0-00075, Artificial Intelligence Graduate School Program, KAIST; No.2022-0-00077, AI Technology Development for Commonsense Extraction, Reasoning, and Inference from Heterogeneous Data, No.2022-0-00984, Development of Artificial Intelligence Technology for Personalized Plug-and-Play Explanation and Verification of Explanation), a National Research Foundation of Korea (NRF) grant (NRF-2021H1D3A2A03103683) funded by the Korean government (MSIT), the KAIST-NAVER hypercreative AI center, and Samsung Electronics Co., Ltd (IO220816-02015-01). Shao-Hua Sun was supported by the Yushan Fellow Program by the Taiwan Ministry of Education and National Taiwan University.

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

# Appendix

---

**Algorithm 2** BOSS Algorithm

---

**Require:** Dataset $\mathcal{D}^L$ w/ language labels, LLM, Skill Library $Z$, Time limit $T$, max chain length $M$
 1: Pre-train policy $\pi(a|s, z)$, value function $V(s, z)$ on $\mathcal{D}^L$ with offline RL.     ▷ Section 3.1
 2: **while** not converged **do**
 3:     SKILLBOOTSTRAPPING(V, Z, LLM, $\pi$, $\mathcal{D}^L$, $M$, $T$)     ▷ Section 3.2
 4:
 5: **procedure** SKILLBOOTSTRAPPING(V, Z, LLM, $\pi$, $\mathcal{D}^L$, $M$, $T$)
 6:     $s_1 \leftarrow$ Reset environment
 7:     `RolloutData` $\leftarrow$ []
 8:     $z \leftarrow$ sample from discrete distribution with probs $\left[V(s, z_1), V(s, z_2), ..., V(s, z_{|Z|})\right]$.
 9:     $i \leftarrow 0$
10:     `Success` $\leftarrow$ True
11:     **while** $i < M$ and `Success` **do**     ▷ If a rollout fails, break the loop.
12:         $i \leftarrow i + 1$
13:         (`Success`, $\tau$) $\leftarrow$ Rollout $\pi(\cdot|s, z)$ in Environment for at most $T$ steps.
14:         Add $\tau$ to `RolloutData`
15:         **if** `Success` **then**
16:             $z \leftarrow$ SAMPLENEXTSKILL(LLM, `RolloutData`, $Z$)
17:     UPDATEBUFFERANDSKILLREPERTOIRE($\mathcal{D}^L$, `RolloutData`, LLM)
18:     Train $\pi$, $V$ on $\mathcal{D}^L$ with offline RL.
19:
20: **procedure** SAMPLENEXTSKILL(LLM, `RolloutData`, $Z$)
21:     `AllSkills` $\leftarrow$ extract all skill annotations from $Z$.
22:     `SkillChain` $\leftarrow$ extract executed primitive skills from `RolloutData`.
23:     `Prompt` $\leftarrow$ construct prompt from `AllSkills`, `SkillChain`.     ▷ Prompt in Figure 8.
24:     $([\hat{z}_1, ..., \hat{z}_N], [p_1, ..., p_N]) \leftarrow$ Sample $N$ text generations from LLM(`Prompt`) with average token probabilities $p_1, ..., p_N$.
25:     Find closest match in $Z$ to each of $\hat{z}_1, ..., \hat{z}_N$ in embedding space     ▷ Embedding model: `all-mpnet-base-v2` from Reimers and Gurevych [49].
26:     $z \leftarrow$ sample the matches in $Z$ from categorical distribution with parameters $p_1, ..., p_N$.
27:     **return** $z$
28:
29: **procedure** UPDATEBUFFERANDSKILLREPERTOIRE($\mathcal{D}^L$, `RolloutData`, $Z$, LLM)     ▷ See Appendix B.3 for details.
30:     $\tau_1, ..., \tau_k \leftarrow$ extract primitive skill trajectories from `RolloutData`.
31:     **for** $\tau_i$ in $\tau_1, ..., \tau_k$ **do**
32:         $\mathcal{D}^L \leftarrow \mathcal{D}^L \cup \{\tau_{i, z_i}\}$     ▷ Add trajectory to $\mathcal{D}^L$ with annotation $z_i$.
33:     $\tau_{1:k} \leftarrow$ concatenate all trajectories together
34:     $z_{LLM, 1:k} \leftarrow LLM(\tau_{1:k})$ assign name by asking LLM summarize annotations of $\tau_{1:k}$.     ▷ See Appendix A.2 for prompt.
35:     $z_{concat, 1:k} \leftarrow$ "$\{z_1\}.\{z_2\}...\{z_k\}$.'     ▷ Assign another label for the trajectory by concatenating primitive skill annotations.
36:     $\mathcal{D}^L \leftarrow \mathcal{D}^L \cup \{\tau_{LLM, 1:k}, \tau_{concat, 1:k}\}$     ▷ Add to $\mathcal{D}^L$ with annotation $z_{LLM, 1:k}$ and $z_{concat, 1:k}$.
37:     Add $z_{LLM, 1:k}$ as a new skill to $Z$.

---

# A  Dataset and Environment Details

## A.1  ALFRED

### A.1.1  Dataset Details

We base our dataset and environment on the ALFRED benchmark [50]. ALFRED originally contains over 6000 full trajectories collected from an expert planner following a set of 7 high-level tasks with randomly sampled objects (e.g., *"pick an object and heat it"*). Each trajectory has three crowd-sourced annotations, resulting in around 20k distinct language-annotated trajectories. We separate these into only the primitive skill trajectories, resulting in about 141k language-annotated trajectories. Following  Zhang et al. [42], we merge navigation skills (e.g., *"Walk to the bed"*) with the skill immediately following them as these navigation skills make up about half of the dataset, are always performed before another skill, and are difficult to design online RL reward functions for that work across all house floor plans given only the information in the dataset for these skills. After this processing step, the resulting dataset contains 73k language-annotated primitive skill trajectories.

### A.1.2  RL Environment Details

We modified ALFRED similarly to Zhang et al. [42], Pashevich et al. [51] to make it suitable for policy learning by modifying the action space to be fully discrete, with 12 discrete action choices and 82 discrete object types.

Furthermore, we rewrote reward functions for all primitive skill types ("CoolObject", "PickupObject", "PutObject", "HeatObject", "ToggleObject", "SliceObject", "CleanObject") so that rewards can be computed independently of a reference expert trajectory. While our rewards depend on the ground truth primitive skill type, no agents are allowed access to what the underlying true primitive skill type is. All of our reward function are sparse, with 1 for a transition that completes primitive skill and 0 for all other transitions.

### A.1.3  Evaluation Tasks

We generate evaluation tasks by randomly sampling 10 tasks each for 4 unseen ALFRED floor plans, resulting in 40 total tasks unseen tasks requiring anywhere from 2-8 primitive skills to complete. The tasks for each floor plan are sampled randomly from the VALID-UNSEEN ALFRED dataset collected in these plans with the specific object arrangements, and we use the high-level task language descriptions collected by humans for ALFRED as our task descriptions for language-conditioned zero-shot evaluation. See Figure 7 for a histogram of task lengths.

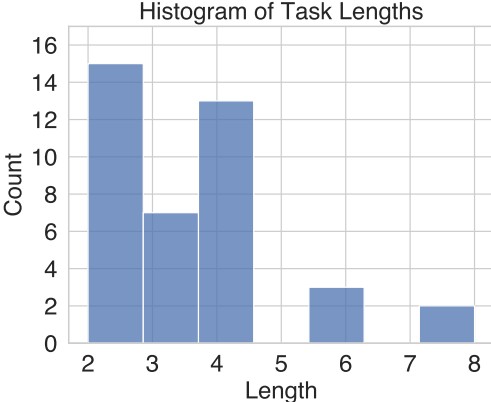

Figure 7: Task lengths regarding the number of primitive skills needed to chain together to solve the task.

Examples of common household tasks and their descriptions:
Task Steps: 1. Pick up the keys on the center table. 2. Put the keys in the box. 3. Pick up the box with keys. 4. Put the box with keys on the sofa close to the newspaper.
Task: Put the box with keys on the sofa.

Task Steps: 1. Pick up the knife from in front of the tomato. 2. Cut the lettuce on the counter. 3. Set the knife down on the counter in front of the toaster. 4. Pick up a slice of the lettuce from the counter. 5. Put the lettuce slice in the refrigerator. take the lettuce slice out of the refrigerator. 6. Set the lettuce slice on the counter in front of the toaster.
Task: Put a cooled slice of lettuce on the counter.

Task Steps: 1. Pick up the book on the table, in front of the chair. 2. Place the book on the left cushion of the couch.
Task: Put a book on the couch.

Task Steps: 1. Pick up the fork from the table. 2. Put the fork in the sink and fill the sink with water, then empty the water from the sink and remove the fork. 3. Put the fork in the drawer.
Task: Put the cleaned fork in a drawer.

Task Steps: 1. Take the box of tissues from the makeup vanity. 2. Put the tissues on the barred rack. 3. Take the box of tissues from the top of the toilet. 4. Put the tissues on the barred rack.
Task: Put the box of tissues on the barred rack.

Task Steps: 1. Pick up the glass from the sink. 2. Heat the glass in the microwave. 3. Put the glass on the wooden rack.
Task: Put a heated glass on the wooden rack.

Task Steps: 1. Pick up the box from the far side of the bed. 2. Hold the box and turn on the lamp.
Tasks: Look at the box under the lamp light.

Predict the next skill correctly by choosing from the following skills: [SKILL 1 IN LIBRARY], [SKILL 2 IN LIBRARY], ...
Task Steps: 1. [SKILL 1 EXECUTED SO FAR] 2. [SKILL 2 EXECUTED SO FAR] ... N. ＿＿＿＿

Figure 8: Prompt for the LLM for next skill proposal (Section 3.2). Text is generated after listing out all skills completed so far.

## A.2   Language Model Prompts

We use two prompts when using the LLM for two different purposes. The main purpose of the LLM is to propose a distribution over next skills to chain with currently executed skills during skill bootstrapping (Section 3.2). Thus, we pass skills in the given skill library $Z$ into the prompt and ask it to predict the next skill. We also include a fixed set of 7 in-context examples from a random sample of different tasks from the ALFRED training dataset. The prompt for bootstrapping is shown in Figure 8.

We also generate summaries (see Section 3.2 and appendix Appendix B.3) for *composite* skill annotations with the LLM. These summaries are used to label newly chained longer-horizon skills before adding them back to the skill library. We show the prompt for this in Figure 9.

## B   Training Implementation details and Hyperparameters

We implement IQL [48] as the base offline RL algorithm to pre-train on primitive skill data for all methods, baselines, and ablations, due to its strong offline and finetuning performance on a variety of dense and sparse reward environments.

The IQL policy is trained to maximize the following objective:

$$e^{\beta(Q(s,a)-V(s))} \log \pi(a|s),$$

which performs advantage-weighted regression [60] with an inverse temperature term $\beta$. $Q$ and $V$ are trained on $(s, a, s', r, a')$ tuples from the dataset rather than sampling a policy for $a'$ to mitigate

Instructions: give a high-level description for the following steps describing common household tasks.

Task Steps: 1. Pick up the keys on the center table. 2. Put the keys in the box. 3. Pick up the box with keys. 4. Put the box with keys on the sofa close to the newspaper.
Summary: Put the box with keys on the sofa.

Task Steps: 1. Pick up the knife from in front of the tomato. 2. Cut the lettuce on the counter. 3. Set the knife down on the counter in front of the toaster. 4. Pick up a slice of the lettuce from the counter. 5. Put the lettuce slice in the refrigerator. take the lettuce slice out of the refrigerator. 6. Set the lettuce slice on the counter in front of the toaster.
Summary: Put a cooled slice of lettuce on the counter.

Task Steps: 1. Pick up the book on the table, in front of the chair. 2. Place the book on the left cushion of the couch.
Summary: Put a book on the couch.

Task Steps: 1. Pick up the fork from the table. 2. Put the fork in the sink and fill the sink with water, then empty the water from the sink and remove the fork. 3. Put the fork in the drawer.
Summary: Put the cleaned fork in a drawer.

Task Steps: 1. Take the box of tissues from the makeup vanity. 2. Put the tissues on the barred rack. 3. Take the box of tissues from the top of the toilet. 4. Put the tissues on the barred rack.
Summary: Put the box of tissues on the barred rack.

Task Steps: 1. Pick up the glass from the sink. 2. Heat the glass in the microwave. 3. Put the glass on the wooden rack.
Summary: Put a heated glass on the wooden rack.

Task Steps: 1. Pick up the box from the far side of the bed. 2. Hold the box and turn on the lamp.
Summary: Look at the box under the lamp light.

Task Steps: 1. [SKILL 1] 2. [SKILL 2] 3. [SKILL 3] ...
Summary:

Figure 9: Prompt for the LLM to summarize completed skills into high-level *composite* annotations, following Zhang et al. [42].

issues with critic function overestimation common in offline RL. We detail shared training and implementation details below, with method-specific information and hyperparameters in the following subsections.

## B.1 ALFRED Environment

We implement the same observation and action space as Zhang et al. [42]. Details are listed below.

**Observation space.** The observations given to agents are $300 \times 300$ RGB images. For all methods, we first preprocess these images by sending them through a frozen ResNet-18 encoder [61] pre-trained on ImageNet, resulting in a $512 \times 7 \times 7$ observation.

**Action space.** The agent chooses from 12 discrete low-level actions. There are 5 navigation actions: `MoveAhead`, `RotateRight`, `RotateLeft`, `LookUp`, and `LookDown` and 7 interaction actions: `Pickup`, `Put`, `Open`, `Close`, `ToggleOn`, `ToggleOff`, and `Slice`. For interaction actions the agent additionally selects one of 82 object types to interact with, as defined by Pashevich et al. [51]. In total, the action space consists of $5 + 7 * 82 = 579$ discrete action choices. For all methods, due to the large discrete action space, we perform the same action masking as Zhang et al. [42] to prevent agents from taking actions that are not possible by using ground truth object properties given by the ALFRED simulator for each object in the scene. For example, we do not allow the agent to `Close` objects that aren't closeable or `ToggleOn` objects that can't be turned on.

**Policy and critic networks.** We use the transformer architecture (and hyperparameters) used by Episodic Transformers (ET) [51] for our policy and critic networks. We implement all critics (two $Q$ functions and one $V$) with a shared backbone and separate output heads. Additionally, we use LayerNorms [62] in the MLP critic output heads as recommended by Ball et al. [63]. All networks condition on tokenized representations of input language annotations.

**Hyperparameters.** Hyperparameters were generally selected from tuning the Oracle baseline to work as best as possible, then carried over to all other methods. Shared hyperparameters for all methods (where applicable) for pre-training on primitive skills are listed below. Any unlisted hyperparameters or implementation details are carried over from Pashevich et al. [51]:

| Param | Value |
|---|---|
| Batch Size | 64 |
| # Training Epochs | 150 |
| Learning Rate | 1e-4 |
| Optimizer | AdamW |
| Dropout Rate | 0.1 |
| Weight Decay | 0.1 |
| Discount $\gamma$ | 0.97 |
| Q Update Polyak Averaging Coefficient | 0.005 |
| Policy and Q Update Period | 1 per train iter |
| IQL Advantage Clipping | [0, 100] |
| IQL Advantage Inverse Temperature $\beta$ | 5 |
| IQL Quantile $\tau$ | 0.8 |
| Maximum Observation Context Length | 21 |

When fine-tuning policies (for Oracle, CIC, and BOSS), we keep hyperparameters the same. We fine-tune one policy per floor plan (zero-shot evaluating on 10 tasks in each floor plan) in our AL-FRED task set so that the aggregated results are reported over 4 runs per seed. For methods that use a skill library (BOSS, Saycan, Saycan+P), all available primitive skills across all evaluation tasks in each floor plan compose the starting skill library, resulting in anywhere from 15-40 available skills depending on the floor plan.

Additionally, when finetuning the Oracle baseline along with BOSS and its ablations, we sample old data from the offline dataset and newly collected data at equal proportions in the batch, following suggestions from [63]. We do not do this for CIC when finetuning with its unsupervised RL objective because the language embeddings from the old data are not compatible with the online collected data labeled with CIC-learned skill embeddings. Fine-tuning hyperparameters follow:

| Param | Value |
|---|---|
| # Initial Rollouts | 50 |
| # Training Steps to Env Rollouts Ratio | 15 |
| $\epsilon$ in $\epsilon$-greedy action sampling | 0.05 |
| Discrete action sampling | True |
| # Parallel Rollout Samplers | 10 |

## B.2 Real Robot Environment

The input observation from the environment includes environment RGB input and robot states. The RGB input consists of the third-person view RGB images from a Logitech Pro Webcam C920 cropped to 224×224×3, and wrist view images from an Intel RealSense D435. We use a pretrained R3M [64] model to get the latent representation for each view. The robot states include the robot's end-effector position, velocity, and gripper state. The end-effector position and velocity are two continuous vectors, and the gripper state is a one-hot vector, which presents OPEN, CLOSE, or NOT MOVE. We concatenate the RGB latent representations and robot states together as environment states.

The policy is language conditioned, and we use a pre-trained sentence encoder to encode the language annotation to a 384-dimensional latent vector. The pretrained sentence encoder we use is `all-MiniLM-L12-v2` from the `SentenceTransformers` package [49].

The total state input dimension is 2048 (third-person R3M) + 2048 (wrist R3M) + 15 (robot state input) + 384 (language latent representation) = 4495.

**Action space.** The action space of the robot encompasses the difference in the end effector position between each time step, along with discrete open and close signals for the gripper. These actions are transmitted to the robot with 10HZ and interpreted as desired joint poses using PyBullet's inverse kinematics module.

In line with [65], we adopt the Action Chunking method to train an autoregressive policy. Our policy utilizes an LSTM model to predict the next 15 actions, given the initial observation as input, denoted as $\pi(a_{t:t+15}|s_t)$. Both our Q and Value networks are recurrent as well, estimating rewards on a per-timestep basis for each action in the sequence. Similar to the policy, these networks only have access to the observation preceding the action sequence initiation.

Due to the gripper action space is discrete and imbalanced distributed in the dataset, we reweigh gripper loss inversely proportionally to the number of examples in each class.

### B.3 Additional BOSS Implementation Details

Here we continue discussion of BOSS in detail. In the main text in Section 3.2 we mention that we add learned skills back to the agent's skill repertoire and then train on collected experience gathered from each rollout. Here, we detail exactly how we do that.

**Labeling new composite skills.** Finally, after we have finished attempting a composite skill chain, we need a natural language description for it so we can train the language-conditioned policy on this new composite skill. We ask the LLM to generate high-level task descriptions of the annotations of the two skills the agent has just attempted to chain together like proposed by Zhang et al. [42] for offline policy pre-training. Doing so will allow the agent to learn skills at a higher level of text abstraction, allowing the agent to operate on more natural evaluation task specifications. For example, humans are more likely to ask an agent to "Make coffee" than to say "Get a coffee pod. Put the coffee pod in the machine. Fill it up with water..."

We give the LLM a prompt similar to the one for generating next skills. For example, if our agent has just completed two skills: *"Pick up the spoon"*, *"Put the spoon on the counter"*, we ask the LLM to summarize "1. PICK UP THE SPOON. 2. PUT THE SPOON ON THE COUNTER.", and the LLM can generate *"put a spoon on the counter."* We denote the generated language annotation for this combined skill composed of the annotations of $z^1$ and $z^2$ as $z'$. We then add $z'$ as a new composite skill to $Z$ for the agent to possibly sample from again.

**Training on new skill data.** After the agent has finished a rollout in the environment, it trains on the experience gathered. There are three types of data that we add to the agent's replay buffer from its rollout data:

1. The trajectory of the attempted skill chain which is collected only if the entire first skill is successfully executed (regardless if it is a primitive skill or a chain of them) since only then will another skill be used for chaining. The label for this trajectory is produced by the LLM.

2. The trajectory of the composite skill but with a label generated by concatenating the primitive skill annotations as a sequence of sentences of their language annotations. This trajectory ensures that the agent receives a description for the collected composite trajectory that specifies the exact primitive skills that make it up, in order. This is useful because the LLM-generated high-level skill description may not describe certain steps. Those steps are explicitly spelled out in this new label.

3. Trajectories for all lowest-level primitive skills executed during the rollout. These correspond to the original set of skills the policy was equipped with and will help the policy continue to finetune its ability to execute its original primitive skills.

After the rollout, we add these trajectories to the agent's replay buffer.

**Other details.** When performing skill bootstrapping in the ALFRED environment, we set a max time limit ($T$ in Algorithm 2) for 40 timesteps per primitive skill. For simplicity, we restrict $M$, the max number of skills to chain, to be 2 during skill bootstrapping rollouts. We also restrict the second skill to be chained to only the set of primitive skills so that the agent can only learn new skill chains that are one primitive skill longer than the first sampled skill. Note that this does not restrict the agent from sampling composite skills it has learned during bootstrapping as first skills upon initialization.

One final implementation detail is with respect to how we map LLM next skill proposals to existing skills in the skill library $Z$. We found that pre-trained sentence embedding models generally seem to put great emphasis on the *nouns* of skill annotation sentences in ALFRED, instead of the verb. Therefore, all sentence embeddings models we initially experimented with (up to the 11B parameter model FLAN-T5-XXL [66]) would have a tendency to map LLM generations such as *"Place the apple in the sink"* to skills with *different verbs* as long as the nouns were the same, such as *"Pick up the apple from the sink"*. These skills are clearly very different, so this presented a problem to us initially. To solve this, we settled on using an NLP library[2] to extract the main verb of sentences and then added that same verb as a prefix to each sentence before embedding with the sentence embedding model. For example, *"Place the apple in the sink"* → *"PLACE: Place the apple in the sink."* With this change, the aforementioned issue was addressed in most cases and we could use much smaller sentence embedding models (`all-mpnet-v2` from the SentenceTransformers package [49]).

**Training Time and Hardware Requirements** We perform experiments on a server with 2 AMD EPYC 7763 64-Core Processors, and 8 RTX 3090 GPUs. Pre-training the policies takes around 10 hours with just a single RTX 3090 and 4 CPU threads for parallel dataloading.

Skill bootstrapping experiments require just 1 GPU with sufficient VRAM to run inference with our LLM, along with 4 available CPU threads for parallel dataloading and environment rollouts. In practice, a single RTX 3090 is sufficient for our experiments using LLaMA-13B with 8-bit inference [67] on ALFRED, requiring around 3-5 days of training, mainly due to the speed of the underlying simulator used in ALFRED.

### B.4   CIC Implementation

For fairness in our experimental comparison, we implement CIC [52] by using its objective to train a policy pre-trained on the same dataset as BOSS; thus, the CIC agent is first initialized with a set of sensible behaviors. Since CIC operates on a fixed latent space, we modified the critic and policy architectures so that they operate on fixed-length, 768-dimensional embeddings of language inputs from the same sentence embedding model used for skill bootstrapping [49] instead of on variable length tokenized language representations.

CIC-specific hyperparameters follow:

---

[2] https://github.com/chartbeat-labs/textacy

| Param | Value |
| --- | --- |
| CIC K-means K | 12 |
| CIC K-means avg | True |
| CIC Hidden Dim | 1024 |
| CIC Latent Skill Dim | 768 |
| CIC Temp | 0.5 |
| CIC Skill Projection Layer | True |
| # Timesteps for each skill rollout before reset | 200 |

## B.5 SayCan Implementation

We implement SayCan [12] by combining the prompt from SayCan with ours. We use the same in-context examples except but convert them to a human-robot conversation. All other details are the same, including the LLM that we use in this comparison (LLaMa-13b [55]). The Saycan prompt follows below:

---

Robot: Hi there, I'm a robot operating in a house. Robot: You can ask me to do various tasks and I'll tell you the sequence of actions I would do to accomplish your task.

Human: How would you put the box with keys on the sofa?
Robot: 1. Pick up the keys on the center table. 2. Put the keys in the box. 3. Pick up the box with keys. 4. Put the box with keys on the sofa close to the newspaper.

Human: How would you put a cooled slice of lettuce on the counter?
Robot: 1. Pick up the knife from in front of the tomato. 2. Cut the lettuce on the counter. 3. Set the knife down on the counter in front of the toaster. 4. Pick up a slice of the lettuce from the counter. 5. Put the lettuce slice in the refrigerator. take the lettuce slice out of the refrigerator. 6. Set the lettuce slice on the counter in front of the toaster.

Human: How would you put a book on the couch?
Robot: 1. Pick up the book on the table, in front of the chair. 2. Place the book on the left cushion of the couch.

Human: How would you put the cleaned fork in a drawer?
Robot: 1. Pick up the fork from the table. 2. Put the fork in the sink and fill the sink with water, then empty the water from the sink and remove the fork. 3. Put the fork in the drawer.

Human: How would you put the box of tissues on the barred rack?
Robot: 1. Take the box of tissues from the makeup vanity. 2. Put the tissues on the barred rack. 3. Take the box of tissues from the top of the toilet. 4. Put the tissues on the barred rack.

Human: How would you put a heated glass on the wooden rack?
Robot: 1. Pick up the glass from the sink. 2. Heat the glass in the microwave. 3. Put the glass on the wooden rack.

Human: How would you look at the box under the lamp light?
Robot: 1. Pick up the box from the far side of the bed. 2. Hold the box and turn on the lamp.

Predict the next skill correctly by choosing from the following skills: [SKILL 1 IN LIBRARY], [SKILL 2 IN LIBRARY], ...

Human: How would you [HIGH LEVEL TASK DESCRIPTION]?
Robot: 1. [SKILL 1 EXECUTED SO FAR] 2. [SKILL 2 EXECUTED SO FAR] ... N. ____

---

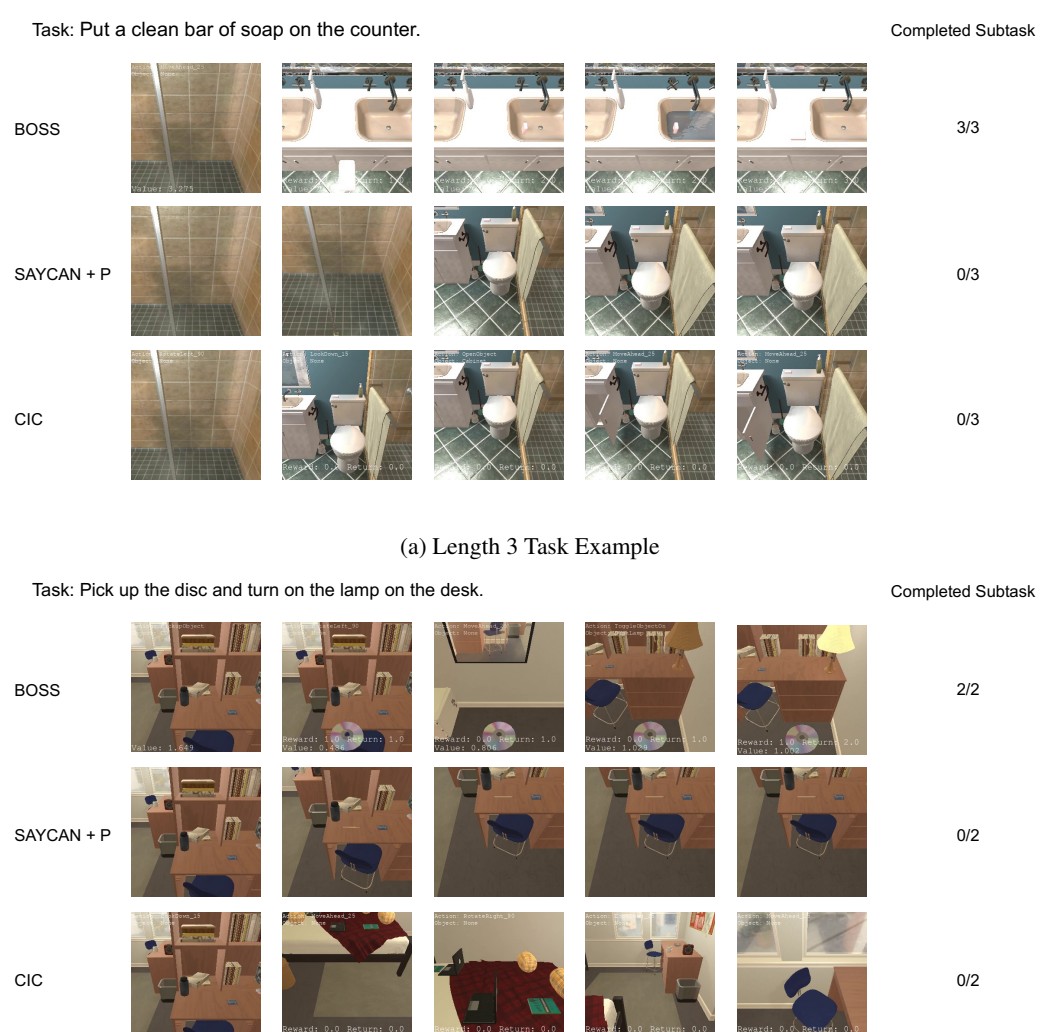

(a) Length 3 Task Example

(b) Length 2 Task Example

Figure 10: Qualitative visualizations of zero-shot evaluation rollouts. See the plans SayCan+P generated for these two tasks at the top of Figure 12.

## B.6 ProgPrompt Implementation

ProgPrompt [14] converts natural language queries to code and executes the code on a real robot. After consulting with the authors, we converted the examples in our prompt to one suitable for ProgPrompt by converting task descriptions into a code representation by converting spaces into underscores, e.g., "Pick up the milk" into `def pick_up_the_milk()`. Then, to translate code commands into commands suitable for our pre-trained policy, we prompt ProgPrompt to output `pick_and_place(object, object)` style code commands that we convert into two separate pick and place natural language commands in the same format as the instructions used for pre-training the policy. We then execute these instructions on the real robot in sequence.

BOSS  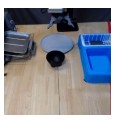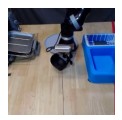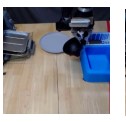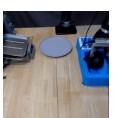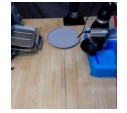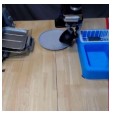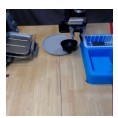  4/4

Figure 11: Example of a BOSS rollout after skill bootstrapping on task 4: "Clean the black bowl and put it in the gray plate." BOSS is able to complete all 4 tasks in this rollout after performing skill bootstrapping.

---

**Task: Put a clean bar of soap on the counter. (Execution Fail)**

GROUND TRUTH

1. Pick up the bar of soap.
2. Put the bar of soap in the sink, turn the water on and then off and then pick up the bar of soap.
3. Put the soap down in between the two sinks.

SAYCAN+P GENERATED PLAN

1. Pick up the bar of soap.

---

**Task: Pick up the disc and turn on the lamp. (Execution Fail)**

GROUND TRUTH

1. Pick up the disc on the desk.
2. Turn on the lamp on the desk.

SAYCAN+P GENERATED PLAN

1. Pick up the disc on the desk.

---

**Task: Examine a bowl by the lamp. (Planning Fail)**

GROUND TRUTH

1. Pick up the bowl on the desk.
2. Turn on the lamp.

SAYCAN+P GENERATED PLAN

1. Pick up the bowl on the desk.
2. Pick up the bowl on the desk.

---

**Task: Put cooked apple slice on a counter. (Planning Fail)**

GROUND TRUTH

1. Pick up the butter knife that is in front of the bowl on the counter.
2. Cut the apple that is in the garbage can into slices.
3. Put the knife in the garbage can.
4. Pick up a slice of apple that is in the garbage can.
5. Put the apple in the microwave and turn it on to cook, remove the cooked apple from the microwave.
6. Put the slice of apple on the counter to the right of the statue.

SAYCAN+P GENERATED PLAN

1. Pick up a slice of apple that is in the garbage can.

Figure 12: Example plans from SayCan+P [12] evaluated on *EVAL_{INSTRUCT}*. SayCan+P errors mainly come from policy execution failures.

# C Additional Results

## C.1 ALFRED Results

**SayCan Performance Analysis.** Here, we analyze the performance of the SayCan baselines in great detail to determine *how* and *why* they perform poorly. SayCan errors occur for two reasons: (1) Planning errors in which the LLM fails to output the correct low-level instruction based on the high level task description, and (2) Policy execution errors in which the policy fails to execute the task correctly, given the correct instruction.

Qualitative examples of BOSS compared to SayCan+P and CIC are shown in Figure 10, where we see that SayCan+P is unable to solve either task. Why is this? The first two plans in Figure 12 correspond to the top two tasks in Figure 10. As we can see, SayCan+P generated the correct first step but the policy failed to execute the skill as SayCan does not fine-tune policies in the environment. While Figure 12 demonstrates that SayCan+P can make partial progress towards certain tasks, it relies on zero-shot LLM execution over fixed policies and therefore does not fine-tune the policies in the environment nor learn to *chain* them together so that the policy is robust enough to transition between skills in new settings.

We analyze the overall proportions of policy execution failures and planning failures for the SayCan baselines in Table 4. We see that SayCan mostly fails at planning (57.5% of the time) while SayCan+P, using BOSS' skill proposal mechanism, mainly fails at execution. Meanwhile, SayCan+PF performs similarly to SayCan+P, indicating that naïve fine-tuning does not greatly improve the success rate of the final plans.

Table 4: Comparison of SayCan and SayCan+P Methods

| Method | Failure Rate (%) | |
|---|---|---|
| | Planning | Execution |
| SayCan | 57.5 | 42.5 |
| SayCan+P | 4.2 | 95.8 |
| SayCan+PF | 5.0 | 95.0 |

**SayCan+BOSS.** Here, we test one more method which combines the advantages of top-down LLM planning methods like SayCan with BOSS' ability to enable agents to learn how to *chain together skills* directly in the target environment. We evaluate SayCan+BOSS, a baseline which breaks down high-level task instructions using SayCan and then issues the commands to BOSS agents after they have performed skill bootstrapping in the target environments. Results in the below table indicate that this baseline performs much better than BOSSalone, indicating that BOSS' LLM-guided skill bootstrapping enables it to learn robust policies that can even be combined with planners to better execute the given plans than naïve fine-tuning with SayCan+PF. Yet if there is no powerful LLM available at test time, BOSS alone still performs very well.

| Method | Evaluation Task Length | | | Average | |
|---|---|---|---|---|---|
| | Length 2 | Length 3 | Length 4 | Return | Success |
| No Bootstrap | 0.03 +- 0.02 | 0.05 +- 0.07 | 0.08 +- 0.09 | 0.03 +- 0.01 | 0.00 +- 0.00 |
| CIC [52] | 0.02 +- 0.02 | 0.25 +- 0.08 | 0.18 +- 0.07 | 0.11 +- 0.01 | 0.00 +- 0.00 |
| SayCan [12] | 0.06 +- 0.02 | 0.14 +- 0.00 | 0.10 +- 0.12 | 0.06 +- 0.00 | 0.00 +- 0.00 |
| SayCan + P | 0.08 +- 0.04 | 0.28 +- 0.00 | 0.20 +- 0.15 | 0.12 +- 0.01 | 0.00 +- 0.00 |
| SayCan + PF | 0.64 +- 0.06 | 0.49 +- 0.20 | 0.59 +- 0.02 | 0.57 +- 0.05 | 0.00 +- 0.00 |
| BOSS (ours) | 0.47 +- 0.12 | 0.59 +- 0.13 | 0.81 +- 0.13 | 0.57 +- 0.06 | 0.57 +- 0.14 |
| SayCan+BOSS (ours) | **0.84 +- 0.16** | **0.87 +- 0.18** | **0.96 +- 0.13** | **0.84 +- 0.06** | **1.02 +- 0.12** |

## C.2 Real Robot Results

We evaluate on 4 tasks, detailed below, in the environment setup shown in Figure 11.

1. Clean the black bowl (length 2): (1) Pick up the black bowl, (2) put it in the sink.

2. Put the black bowl to the dish rack (length 2): (1) Pick up the black bowl, (2) put it in the dish rack.

3. Clean the black bowl and put it in the dish rack (length 4): (1) Pick up the black bowl, (2) put it in the sink, (3) pick up the black bowl, (4) put it in the dish rack.

Table 5: Full returns and success rates for real robot evaluation comparisons.

| Task | ProgPrompt return | ProgPrompt success rate | BOSS return | BOSS success rate |
|------|-------------------|-------------------------|-------------|-------------------|
| 1 | $1.6 \pm 0.80$ | 0.8 | $1.6 \pm 0.8$ | 0.8 |
| 2 | $1.0 \pm 1.00$ | 0.5 | $0.8 \pm 0.75$ | 0.2 |
| 3 | $0.9 \pm 0.78$ | 0.0 | $1.7 \pm 1.1$ | 0.1 |
| 4 | $2.0 \pm 1.2$ | 0.0 | $2.2 \pm 0.98$ | 0.2 |

4. Clean the black bowl and put it in the gray plate (length 4): (2) pick up the black bowl, (2) put it in the sink, (3) pick up the black bowl, (4) put it in the plate.

We report full results in Table 5.

