# OpenReview forum: "Bootstrap Your Own Skills: Learning to Solve New Tasks with Large Language Model Guidance"
_robot-learning.org/CoRL/2023/Conference — CoRL 2023 Oral_

### Official Review · Reviewer_MjyV · 2023-07-14

**Confidence:** 3
**Originality:** Very Good
**Technical Quality:** Very Good
**Clarity Of Presentation:** Good
**Impact:** 4

**Recommendation:**

Weak Accept: I recommend accepting the paper, but will not argue for my recommendation if the majority of other reviewers have a different opinion.

**Review:**

Strength

1.	The method is novel
2.	The performance of the method is good
3.	The research direction is important

Weakness

Many essential details that are critical for understanding the method are only explained in the appendix. I believe the main paper should at least briefly describe how to add online trajectories into the replay buffer. I was super confused about this point until I read L616-L628 in the appendix. The main paper should also briefly mention how they combine offline and online data in the online finetuning stage, which is elaborated in L566 in the appendix.


**Quality Of The Limitations Section:**

Limitations are addressed clearly

**Questions For Rebuttal:**

1.	In Tab.1, why the performance keeps increasing when the task length is longer? For BOSS, the performance of Length 2 is 0.48, but for Length 4 is 0.71.
2.	In Tab.3, BOSS-Rand, which doesn’t use LLM at all, outperforms SayCan in Tab.1 by a large margin. Why does this happen?

Mirror
Figure 1 is not referred to in the paper.


**Robotics Focus:**

Sufficient demonstration on hardware

**Summary Of Paper:**

This paper proposes an LLM-based method (named BOSS) that chains skills into new composite skills for better performance in long-horizon tasks. Specifically, each existing skill has a corresponding language description. An LLM is used to infer the next reasonable skill. The complete episodes consist of a chain of skills defined as a new composite skill, and the LLM is used to generate the description of the new skill. A language-conditioned policy is trained on offline and online data. Experiment results show that BOSS outperforms baselines including CIC, SayCan, and ProgPrompt in both ALFRED Household simulator and real robots by a large margin.

**Summary Of Recommendation:**

In general, I think this paper introduces a novel and effective way to utilize LLM’s planning ability for better performance in long-horizon tasks. Based on its novelty and performance, I think this paper deserves a presentation at the conference.

---

### Official Review · Reviewer_e5v5 · 2023-07-15

**Confidence:** 4
**Originality:** Good
**Technical Quality:** Very Good
**Clarity Of Presentation:** Very Good
**Impact:** 3

**Recommendation:**

Strong Accept: I recommend accepting the paper and will argue for my recommendation even if other reviewers hold a different opinion.

**Review:**

Strengths:

1. Approach: Proposes a principled approach to incorporate LLM guidance in unsupervised RL-based skill learning.

2.  Good set of Experiments: Exhaustive baselines and ablations have been devised. The results demonstrate utility of the approach.

Weaknesses:

1. The analysis of failure modes of the baselines should be improved. Specifically, the reason for SayCan and SayCan+P baselines performing so poorly as compared to BOSS even for tasks of length 2 should be mentioned more explicitly. My understanding is that SayCan would be able to "decompose" the instruction (at least for length 2) into the sequence of primitive skills correctly, but the scores are low because the intermediate state in-between two skill executions is not seen during pre-training. This should be mentioned more clearly in the paper, possibly with a visual demonstration emphasizing the failure. Currently lines 253-255 handle this but I believe more clarity is warranted.

     1.1 In this context, I propose another baseline: SayCan+P+"primitive skills pre-trained on all floor plans (even those used in evaluation)". Comparison with this baseline would demonstrate the efficacy of LLM-guided skill composition v/s LLM-based planning modulo policy execution errors.

2. Line 545-547 in the appendix mention that "we do not allow the agent to Close objects that aren’t closeable.....". How is it determined if a particular object is closeable during test time?

3. In the supplementary webpage that contains videos, the language instruction for the last demonstration is missing.

4. Another missing detail is the total training time of the system.

**Quality Of The Limitations Section:**

Limitations are addressed clearly

**Questions For Rebuttal:**

1. Please provide a comparison of your approach (LLM-guided skill composition) v/s LLM-based planning (SayCan+P), modulo policy execution errors.

2. What is the total training time and hardware requirements for training the system?

3.  Line 545-547 in the appendix mention that "we do not allow the agent to Close objects that aren’t closeable.....". How is this performed?

**Robotics Focus:**

Sufficient demonstration on hardware

**Summary Of Paper:**

The paper tackles the problem of learning complex skills as composition of pre-trained primitive skills. The core idea of the paper is to leverage LLMs to prune the exponentially large space of skill compositions into the space of compositions that are plausible in the real world. This, along with the idea of bootstrapping skills as they are acquired, enables tractable learning.

Given the first skill from an initial set of pre-trained primitive skills (trained via language-conditioned RL), an LLM is prompted to get a physically plausible next primitive skill. The combined trajectory for the skill-chain along with a succinct language instruction for the chain (using another LLM call) is added to the training data for the RL algorithm.

Extensive baselines, ablations and real hardware experiments are performed.

**Summary Of Recommendation:**

I have not given a lower rating because:

1. Good set of experiments have been performed. The approach is sufficiently novel.
2. Leveraging LLMs and other foundation models to provide strong priors for learning long-range behaviours is a promising direction.

I have not given a higher rating because:
1. Additional baseline experiments would provide more clarity.
2. I dont forsee a major advancement in any particular area arising from this work.

---

### Official Review · Reviewer_CC9W · 2023-07-22

**Confidence:** 4
**Originality:** Very Good
**Technical Quality:** Good
**Clarity Of Presentation:** Very Good
**Impact:** 3

**Recommendation:**

Weak Accept: I recommend accepting the paper, but will not argue for my recommendation if the majority of other reviewers have a different opinion.

**Review:**

The paper presents an idea that is in nice contrast to recent approaches that use LLMs as top-down planners. I think it is worthwhile to explore this bottom-up direction of building the skill library. Though it's not a completely new idea to use LLMs as guidance for sampling new tasks, this paper offers a convincing explanation to how it differs from prior works.

The paper is very well written. The evaluation is thorough and there are impressive demonstrations on a real robot manipulator. However, there are a few points I'm not fully convinced on.

First, the performance of SayCan is surprisingly bad, and I do not understand the explanations provided. In Figure 10, it looks like SayCan found the soap but failed to pick it up, which matches the description on line 258 that it got stuck while "attempting to manipulate objects." This means SayCan selected the correct skill, but the pre-trained policy failed during execution. However, according to line 218, this is the same pre-trained policy as in BOSS. I wonder how many of the failure cases can be attributed to SayCan or SayCan+P making the wrong plan, and how many are policy execution failures. I think further analysis is needed here to make sure the performance improvement of BOSS is not mostly due to having seen more trajectories to fine-tune the primitive skills. I would also be interested to see an evaluation of the final primitive skills trained by BOSS with the step-by-step instructions generated by SayCan+P.

Further, it feels like a missed opportunity to not combine the top-down planning and the bottom-up skill expansion in some way. Based on the last example in Figure 5, the LLM could propose unrelated chains of skills and give an incorrect summary. Perhaps it's better to give the high-level task description and the skill chain together. I also wonder if eventually the large number of meaningless skill chains will cause catastrophic forgetting.


**Quality Of The Limitations Section:**

Limitations are addressed clearly

**Questions For Rebuttal:**

- Can you provide more details on the failure cases of SayCan+P in Figure 10? What skills were selected by the LLM, and where it failed during execution?
- Is the state of the environment provided to the LLM? When it proposes "pick up the keys on the table" how does it know there are keys on the table?
- Can you explain how invalid skill proposals from the LLM are handled? For example, "place keys in the safe" when the robot has not picked up keys.


**Robotics Focus:**

Sufficient demonstration on hardware

**Summary Of Paper:**

This paper presents an approach called BOSS which learns long-horizon tasks by practicing chains of primitive skills guided by large language models. This approach prompts an LLM to propose the next steps from the current skill library, executes the skills, and summarizes the composite skill using the same LLM. From this mostly self-supervised procedure, the robot can quickly acquire a large library of language-conditioned skills. BOSS is evaluated in the ALFRED benchmark and real-robot manipulation tasks. The results show that BOSS outperforms prior baselines by achieving higher success rates on language-specified tasks that require sequencing of multiple primitive skills.


**Summary Of Recommendation:**

I recommend accepting the paper because this paper proposes a very interesting and relevant idea and executes it well, but there is room for improvement in the presentation of design choices and the analysis of the evaluation results.

---

### Author Response · Authors · 2023-08-13
**BOSS Rebuttal Overview**

We thank all of the reviewers for their constructive comments and thorough, positive reviews! We are happy reviewers found our paper well written (CC9W), the experiments and evaluation thorough (CC9W, e5v5), and the method novel (MjyV) and principled (e5v5).

We have addressed each reviewer’s concerns individually. Following their suggestions, we have made the following **major changes** to our paper:

- Added a **new baseline** that directly fine-tunes SayCan’s policies on the target environments before zero-shot evaluation. Our method still achieves a **57% success rate**, while this baseline achieves **0%** (CC9W, e5v5).
- Performed extensive analysis of why SayCan’s performance is poor on our tasks: **added appendix Section C.1, Fig. 12, and Table 4** extensively analyzing SayCan’s performance and visualizing some example failed plans made by SayCan (CC9W, e5v5).
- *Clarified* many details (MjyV, CC9W, e5v5) by adding new text and moving text from the appendix to the main paper.

---

### Author Response · Authors · 2023-08-14
**1 Day Left: BOSS Rebuttal Discussion Reminder**

This is a gentle reminder to our reviewers that the discussion period ends tomorrow.

We thank all of the reviewers for their detailed reviews, and we would love to have the chance to discuss further with all reviewers.

We have introduced a much-requested **new baseline, fine-tuned SayCan**, which BOSS greatly outperforms in terms of success rate.

We have also added extensive analysis on SayCan's performance to the rebuttal PDF, and made clarification changes to the paper.

Thanks!

BOSS Authors

---

### Decision · Program_Chairs · 2023-08-30

**Decision:**

Accept (Oral)

**Comment:**

The paper proposes an approach using the long-horizon planning and language capabilities of LLMs to guide the learning of a growing skill library in an open-ended fashion. The reviewers agree on the novelty of the approach, the provided experiments, and the general research direction that is promising.
However, there was also the shared opinion regarding the comparison result against SayCan and how performance is evaluated. The authors provided an extensive rebuttal, addressing the main concerns and questions. All reviewers agree on the quality of the paper and to accept it.